# Towards Unifying Interpretability and Control: Evaluation via Intervention

## Abstract

With the growing complexity and capability of large language models, a need to understand model reasoning has emerged, often motivated by an underlying goal of controlling and aligning models. While numerous interpretability and steering methods have been proposed as solutions, they are typically designed either for understanding or for control, seldom addressing both. Additionally, the lack of standardized applications, motivations, and evaluation metrics makes it difficult to assess methods' practical utility and efficacy. To address the aforementioned issues, we argue that intervention is a fundamental goal of interpretability and introduce success criteria to evaluate how well methods can control model behavior through interventions. To evaluate existing methods for this ability, we unify and extend four popular interpretability methods—sparse autoencoders, logit lens, tuned lens, and probing—into an abstract encoder-decoder framework, enabling interventions on interpretable features that can be mapped back to latent representations to control model outputs. We introduce two new evaluation metrics: intervention success rate and coherence-intervention tradeoff, designed to measure the accuracy of explanations and their utility in controlling model behavior. Our findings reveal that (1) while current methods allow for intervention, their effectiveness is inconsistent across features and models, (2) lens-based methods outperform SAEs and probes in achieving simple, concrete interventions, and (3) mechanistic interventions often compromise model coherence, underperforming simpler alternatives, such as prompting, and highlighting a critical shortcoming of current interpretability approaches in applications requiring control.

## 1 Introduction

As large language models (LLMs) have become more capable and complex, there has emerged a need to better understand and control these models to ensure their outputs are safe and human-aligned. Many interpretability methods aim to address this problem by analyzing model representations, attempting to understand their underlying computational and reasoning processes in order to ultimately control model behaviour. While many of these methods, and interpretability as a field more broadly, claim control and intervention as abstract goals and present compelling qualitative results demonstrating that intervention may be possible in certain cases (for example, Anthropic's Golden Gate Claude Anthropic; Templeton et al. (2024)), the link between interpretation and intervention is tenuous in practice, and many methods are not explicitly tailored for both. Furthermore, even fewer are thoroughly and systematically evaluated for the ability to control model outputs beyond qualitative examples. We believe the reason for this is threefold. First, interpretability methods produce explanations in disparate feature spaces, such as token vocabulary, probe predictions, or learned auto-interpreted features, hindering comparisons across methods. Second, there exists a "predict/control discrepancy" (Wattenberg & Viégas, 2024), where the features identified by interpretability methods for *predicting* behavior are not the same as those used for *steering* it. Third, there do not exist standard systematic benchmarks to measure intervention success.

In this work, we view intervention as a fundamental goal of interpretability, and propose to measure both the correctness and the utility of interpretability methods by their ability to successfully edit model behaviour. In particular, we focus on sparse autoencoders (Cunningham et al., 2023), Logit Lens (nostalgebraist; Dar et al., 2023), Tuned Lens (Cunningham et al., 2023; Rajamanoharan et al., 2024; Templeton et al., 2024; Bricken et al., 2023; Gao et al., 2024), and linear probing

(Alain, 2016; Belinkov & Glass, 2019; Belinkov, 2022), and benchmark them with steering vectors and prompting as baselines for intervention. In order to enable comparison across these various methods, we first unify and extend the methods as instances of an abstract encoder-decoder framework, where each method encodes uninterpretable latent representations of language models into human-interpretable features and the decoder of the framework inverts this mapping, allowing us to reconstruct a latent representation from the features. Under this abstract framework, we can intervene on the interpretable feature activations generated by each method and decode them into latent counterfactuals, which produce counterfactual outputs corresponding to the desired intervention.

The unifying feature interpretation and intervention framework allows us to propose two standard metrics for evaluating mechanistic interpretability methods: (1) intervention success rate, which measures how well intervening on an interpretable feature causally results in the desired behavior in the model outputs, and (2) coherence-intervention tradeoff, which measures how well the causal interventions succeed without damaging the coherence of the model's outputs. We evaluate Logit Lens, Tuned Lens, sparse autoencoders, and linear probes for these metrics on GPT2-small, Gemma2-2b, Llama2-7b, and Llama3-8b, comparing them to simpler but uninterpretable baselines of steering vectors and prompting. Our results show that while existing methods allow for intervention, their effectiveness is inconsistent across features and models. Furthermore, lens-based methods outperform all other methods, including sparse autoencoders, for simple, concrete features, likely due to the spurious correlation learned by probes and steering vectors and the high error rate in SAE feature labeling pipelines. We further show that intervention often comes at the cost of model output coherence, underperforming simple prompting baselines, presenting a critical shortcoming of existing methods in real-world applications that require control and intervention. We conclude this work with some case studies of intervention on complex and safety-relevant features, along with detailed takeaways about the strengths and weaknesses of each method, including discussion of which methods are optimal for specific intervention topics, which are best to use out of the box, and which hold the most promise for future development.

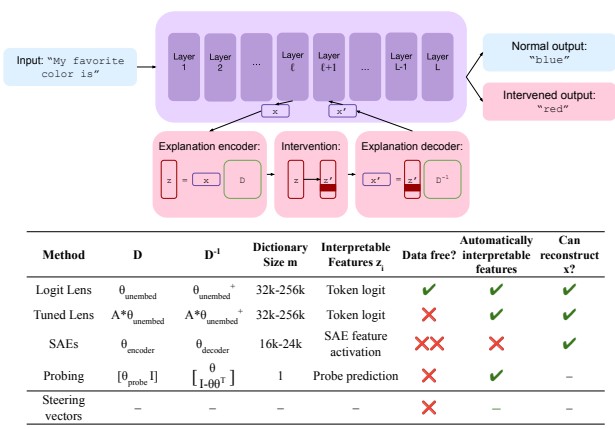

Figure 1: Our proposed intervention framework, which encodes model latent representations, $x$, into human-interpretable features, $z = xD$, that can then be perturbed to $z'$ and decoded back into counterfactual latent representations, $\hat{x}'$.

Our main contributions include:

- In Section 3.1, we present a unifying framework for four popular interpretability methods: sparse autoencoders, logit lens, tuned lens, and probing. To faciliate this, we extend logit lens and tuned lens methods with decoders to allow for intervention.
- In Section 3.2, we propose two evaluation metrics for encoder-decoder interpretability methods, namely (1) intervention success rate and (2) the coherence-intervention tradeoff to evaluate the ability of interpretability methods to control model behavior, and design an open-ended prompt dataset for benchmarking interpretability methods.
- In Section 4, we perform experimental analysis on GPT-2, Gemma2-2b, Llama2-7b, and Llama3-8b, and present detailed takeaways comparing interpretability- and control-based methods.

Overall, this paper takes a key step in establishing systematic benchmarks for mechanistic interpretability methods, making progress towards a previously stated open problem for the field (Mueller et al., 2024).[1]

---

[1] All code and data will be released upon acceptance.

## 2 RELATED WORK

**Mechanistic Interpretability.** Existing work in mechanistic interpretability broadly falls into two categories: activation patching and interpreting hidden representations. Activation patching utilizes carefully constructed counterfactual representations to study which neurons or activations play key roles in model computation, ideally localizing specific information to individual layers, token positions, and paths in the model (Geiger et al., 2021; Vig et al., 2020). However, recent work points to key limitations of patching, particularly with respect to real-world utility in downstream applications such as model editing (Hase et al., 2024; Zhang & Nanda, 2023). As such, we focus primarily on methods for inspecting hidden representations, of which probes are the most commonly used (Alain, 2016; Belinkov & Glass, 2019; Belinkov, 2022). Other methods such as Logit lens (nostalgebraist; Dar et al., 2023) project intermediate representation into the token vocabulary space, with Belrose et al. (2023); Din et al. (2023); Geva et al. (2022) building and improving upon these early-decoding strategies. Ghandeharioun et al. (2024a) unifies most of these methods into an abstracted framework for inspecting model computation. More recently, sparse autoencoders and dictionary learning have been explored as a solution to the uninterpretability of model neurons, particularly due to issues with polysemanticity and superposition (Elhage et al.; Bricken et al., 2023; Cunningham et al., 2023; Bhalla et al., 2024; Gao et al., 2024; Templeton et al., 2024; Karvonen et al., 2024).

**Evaluation.** Due to the recency of the field, standard evaluation metrics across interpretability methods have not yet been established, and similar to the broader interpretability field, evaluation is frequently ad-hoc and primarily qualitative in nature, with recent works pointing to the need for more causal evaluation (Mueller et al., 2024; Saphra & Wiegreffe, 2024). With regards to quantitative metrics, in (Gao et al., 2024; Templeton et al., 2024; Makelov et al., 2024), sparse autoencoders are evaluated for reconstruction error, recovery of supervised or known features, activation precision, and the effects of ablation; however, none of these metrics measure the correctness of explanations or usefulness for control. Independent of our work, Wu et al. (2025) also propose a benchmark for steering methods, AxBench, to assess whether steering is a viable alternative to existing model control techniques, finding similar results to ours. Different from them, we consider additional lens-based interpretability methods and explore the extent to which intervention is possible without output degradation, for both simple and safety-relevant interventions.

**Causal Intervention.** Previous literature on probing frequently evaluates learned probes and features through intervention to ensure causality and correctness, as done by Li et al. (2022); Chen et al. (2024); Hernandez et al. (2023b;a); Marks & Tegmark (2023). The interventions performed for measuring causality are similar to those used to perform model "steering" (Rimsky et al., 2023; Panickssery et al., 2024; Ghandeharioun et al., 2024b) and should ideally produce the same effect but with the added claim of interpretability. Geiger et al. (2024) unify many interpretability methods and steering through causal abstraction but do not extend or evaluate these methods for control. Mueller et al. (2024); Belrose et al. (2023); Chan et al. (2022); Olah et al. (2020) consider causal intervention as a tool for assessing explanation faithfulness; however, these works often do not compare between methods and do not consider intervention as a means for control, providing no exploration of the quality of the intervened outputs or their utility in application. Templeton et al. (2024) on the other hand provides a qualitative demonstration of intervention via their 'Golden Gate Claude' but do not systematically measure or compare against other interpretability methods. Different from these works, our work aims to adapt and evaluate existing methods (notably, lens-based methods and SAEs) originally proposed as model inspection tools, for intervention.

## 3 METHOD

In this section, we first introduce a unifying framework for four common mechanistic interpretability methods: sparse autoencoders, Logit Lens, Tuned Lens, and probing, along with modifications to these methods that permit principled intervention on representations. We then propose evaluation metrics for (1) testing the correctness of explanations via intervention and (2) the usefulness of these methods for steering and editing representations and model outputs.

### 3.1 UNIFYING INTERVENTION FRAMEWORK

**Latent vectors to interpretable features.** The central aspect of most interpretability work is the ability to translate model computation into human-interpretable features, whether the computation be latent directions, neurons, components, reasoning processes, etc. Many works aiming to explain

LLMs focus particularly on hidden representations, where the mapping between high-dimensional dense embeddings and human-interpretable features is modeled through a (mostly) linear dictionary projection or affine function:

$$z = f(x) = \sigma(x \cdot D)$$
$$\hat{x} = g(z) \approx f^{-1}(z) = z \cdot D^{-1}$$
$$z' = \text{Edit}(z), \quad \hat{x}' = g(z')$$

where each $z_i$ is a feature activation, each $i$ in $D$ corresponds to a human-interpretable feature, and $\sigma$ is an activation function that is frequently the identity. In the case of **sparse autoencoders**, $D$ is a learned, overcomplete dictionary, with 16k - 65k features for small models (up to 16M for large models), and $\sigma$ is a ReLU, JumpReLU, or ReLU + top-k activation function. Given that SAE features are learned, they are not immediately interpretable and must be labelled by humans or strong LLMs after training. For **Logit Lens**, $D$ is simply the language model's unembedding matrix, meaning each feature corresponds to a single token in the vocabulary. For **Tuned Lens**, $D$ is the exact same as Logit Lens but with a learned linear transformation applied. **Linear probes** can be thought of as a learned dictionary with $N = 1$ where $\sigma$ is a sigmoid or softmax activation and the data is labelled. Of all these methods, Logit Lens is the only method that does not require any training data, and sparse autoencoders are the only method that do not produce immediately interpretable features. For a visual summary of this framework, see Figure 1.

**Interpretable features to counterfactual latent vectors.** While producing explanations is straightforward for each method, intervening on model representations using the information provided by explanations is not as simple. Doing so requires defining a reverse mapping from the explanations to the latent representations of the model, which is only explicitly done by sparse autoencoders.

We extend lens-based methods and probing by defining inverse mappings for them as follows. To map **Logit Lens**'s explanations back into the model's latent space, we would ideally apply the inverse of the unembedding matrix to $z$; however, in practice this is often ill-conditioned due to the dimensionality of $D$. As such, we instead use the low-rank pseudoinverse of the unembedding matrix and right-multiply it to the explanation logits. Similarly, for **Tuned Lens**, we model the decoding process through the pseudo-inverse of the Tuned Lens projection applied to the unembedding matrix. Notably, both of these methods only require a simple linear transformation to go back-and-forth between latent vectors and explanations. For **probing**, an inverse mapping $D^{-1}$ is not strictly necessary, as all interventions can be performed directly on $x$ instead of $z$, as done by Chen et al. (2024); however, an inverse mapping can be designed to maximally recover $x$ from $z$, as shown in Figure 1. **Sparse autoencoders** have a well-defined backwards mapping through the SAE decoder, which is frequently linear in practice and often the transpose of the encoder weights.

**Intervening on interpretable features.** Given the above framework, intervention is performed by directly altering the feature activation $z_i$ corresponding to the desired feature $i$ to be edited. While the edited activation $z_i'$ can naively be set to some constant value $\alpha$, the same constant may have drastically varying effects for different tokens and different prompts. As such, to take into account the context of $z$, for Logit Lens, Tuned Lens, and SAEs we set $z_i' = \alpha * \max(z)$. This ensures that the feature $i$ is the most dominant feature in the latent vector for $\alpha > 1$. Decoding $z'$ yields the altered latent representation $\hat{x}' = g(z')$, which accounts for both the error of the explanation method as well as the intervention performed. For probing and steering vectors, $\hat{x}' = x + \alpha * v$, where $v$ is the steering vector or the weights of the linear probe. Note that $\alpha$ is a hyperparameter that must be tuned for each method and model, and thus cannot be used to compare the effects of interventions across methods. In order to do so, we can instead measure the normalized difference between the latent vectors $x$ and $\hat{x}'$, to characterize the strength of the intervention. We also note that $\hat{x}$ and $\hat{x}'$ are not necessarily in-distribution for the language model, but due to the additive nature of the residual stream and the linear representation hypothesis, we believe that such interventions may still be principled in practice (see Park et al. (2023) for more on the linear representation hypothesis and intervention).

## 3.2 EVALUATION ACROSS METHODS AND MODELS

Given the overall lack of standardized evaluation of mechanistic interpretability methods, we intend for this work to serve as a starting point for systematic evaluation by testing methods in simple,

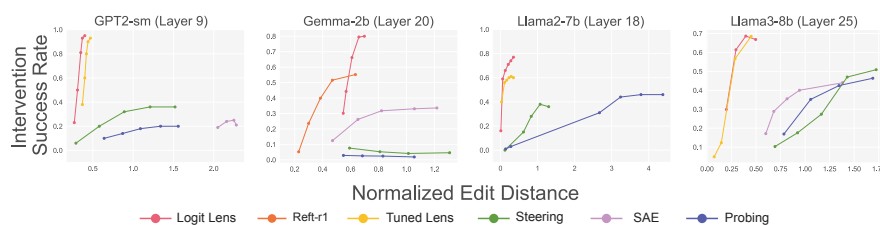

Figure 2: Evaluation of the Intervention Success Rate with respect to edit distance for each method on four models for the simple intervention topics. Note that normalized edit distance is a proxy for intervention strength that is comparable across methods. Logit Lens generally outperforms all other methods.

easy-to-measure contexts. In particular, we think of our evaluations as measuring a kind of upper bound for these methods: in the easiest of settings, how well do existing methods work?

**Explanation Correctness.** We first propose metrics to evaluate the *correctness* of explanations and interventions. More specifically, to test whether a single feature of an explanation $z_i$ is correct, we intervene on that feature to produce $z_i'$ and decode $z'$ to $\hat{x}'$, which should generate text that matches the intervention made to produce $z'$. For example, if feature $i$ encodes the concept "references to Paris," increasing the value of $z_i$ should result in increases to references of Paris in the model's output. From this, we propose a metric of **Intervention Success Rate**, which measures if increasing activation $z_i$ results in the appropriate increase of the feature $i$ in the model's output. To evaluate a continuous relaxation of this, we can also similarly measure the probability assigned to tokens relating to feature $i$. As such, even if the model's output does not directly reflect interventions made to $z_i'$ due to sampling, we can measure if increasing the activation of $i$ results in any change to the model's output at all. We refer to this metric as **Intervened Token Probability**. Importantly, both of these metrics can be thought of as measuring the causal fidelity of the features highlighted by explanations.

**Usefulness of Intervention Methods.** While intervention is a useful method for evaluating the correctness of explanations, it is also a desideratum of its own and a frequent motivation for many explanation methods. For example, methods are often developed for the purpose of de-biasing model outputs or increasing model safety, either by localizing bad behavior or identifying it at inference time, thus allowing for targeted edits to be made. However, a lack of this direct connection between interpretation and model intervention has led to illusory results in prior literature (Hase et al., 2024; Wattenberg & Viégas, 2024). By directly and explicitly measuring how effective interpretability methods are at allowing for targeted intervention or steering, we can avoid such failure cases. Importantly, intervention is only useful if the language model retains its overall performance and still satisfies the purpose of the query as well as the intervention. Thus, we want to evaluate whether interpretability methods can steer model outputs towards feature $i$ without damaging the model. We define **Coherence** as the grammatical correctness, consistency, and relevance to the prompt of the generated text, which can be measured by querying an appropriate oracle, such as a human or strong LLM. Similarly, we can also measure the **Perplexity** of the intervened outputs with respect to a strong language model. In practice, we use Llama3.1-8b for both of these metrics, as it is reasonable sized, high-performing, and open source, allowing for the measurement of perplexity. We compare coherence scores given by Llama3.1-8b to those generated by human raters as well as a rules-based grammar checker to ensure efficacy of our LLM-as-a-judge setup in Table 1.

**An Open-ended Evaluation Dataset.** In order to evaluate these methods to the best of their capabilities, we are interested in assessing their ability to intervene when intervention is straightforward and possible given the prompt. Consider the question "*What is $\int\int sin(3*x)*cos(y)dxdy$?*". Intervening on the model's output for this prompt with a feature related to unicorns is not necessarily intuitive, as there is a correct answer to the prompt that is entirely unrelated to the intervention topic. As such, we want to evaluate these methods on prompts that allow for steering towards a variety of topics or features. To that end, we construct a dataset of 210 prompts related to poetry, travel, nature, journaling prompts, science, the arts, and miscellaneous questions that could plausibly be answered while satisfying a variety of intervention topics. All prompts are open-ended to allow for many potential answers. Some example prompts include "*In ten years, I hope to have accomplished*", "*Check out this haiku I wrote:*", and "*What is your favorite dad joke?*".

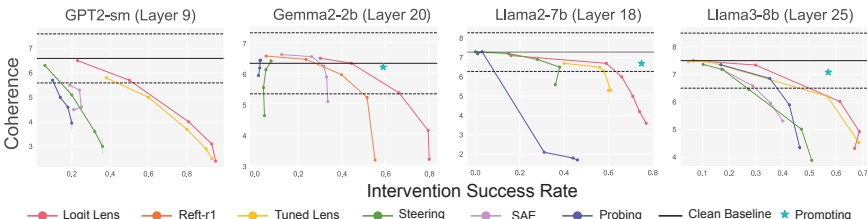

Figure 3: Intervened output coherence measured with respect to intervention success rate. The solid horizontal line shows the mean of coherence scores for the clean model outputs, and the dashed lines show ±1 standard deviation around the mean.

## 4 EXPERIMENTS

In this section, we evaluate the four interpretability methods on our proposed metrics and provide case studies of intervention on more complex concepts. We also present an analysis of the empirical alignment between methods. Additional experiments relating to latent reconstruction error and intervention efficacy across model layers are in Appendix B.1 and B.4.

### 4.1 IMPLEMENTATION DETAILS

**Intervention Topics.** We choose 10 intervention topics that all relate to references to specific words or phrases: {'beauty', 'chess', 'coffee', 'dogs', 'football', 'New York', 'pink', 'San Francisco', 'snow', 'yoga'}, generalizing 'Golden gate Claude'-style interventions. These simple, low-level features are ideal for evaluation through intervention for four key reasons: first, measuring the presence of a word or phrase is much easier than measuring a high-level abstract concept such as sycophancy, second, these features were present in the pretrained and labelled sparse autoencoders we studied, third, the features necessarily exist in the Logit Lens unembedding dictionary, and finally, datasets that are labelled for the presence of these features are very straightforward to collect for generating steering vectors and probes. As such, we can easily compare interventions on these features across all interpretability methods and measure intervention success by checking if the given word/phrase exists in the model's output.

**Steering vectors and probing.** We implement steering vectors with Contrastive Activation Addition (CAA) (Rimsky et al., 2023) with a few simple modifications. Where in CAA, the difference between contrastive pairs is taken only at the last token, we find that averaging across the token dimension and taking the difference between those averages yields much better results. This is due to the fact that in CAA, the only difference between representations occurred in the token position of the answer letter, or the last token; however, in our case the information related to the intervention feature could be present at any token. Example contrastive data pairs were hand-generated by the authors and then used to prompt ChatGPT to create a total of 200 pairs of sentences. All data was verified by the authors and is made available in the accompanying codebase. These contrastive pairs were also used to train the linear probes, using the implementation from Chen et al. (2024). All probes reached train and test accuracies of 100% across all models and intervention topics.

**Sparse autoencoders and supervised dictionaries.** We focus specifically on sparse autoencoders trained to interpret the residual stream of transformer models. We use the SAELens library from Bloom (2024) for GPT2-small and Llama3-8b and the Gemma Scope SAEs (Lieberum et al., 2024) for Gemma2-2b. SAE feature labels were found via Neuronpedia (Lin & Bloom, 2023), which allows users to search through fully trained SAEs and their auto-interpretation labelled features. We also evaluate the Rank-1 Representation Finetuning (ReFT-r1) supervised dictionaries released by Wu et al. (2025), which have features that directly correspond to the SAE features for Gemma2-2b. Note that dictionaries were only released for layer 20 of Gemma2-2b, so we cannot present evaluation for other layers or models.

### 4.2 INTERVENTION SUCCESS ACROSS MODELS

As described in Section 3.2, in order to evaluate the correctness of explanations, we measure the causal effects of intervening on specific features of each explanation. For a given feature or in-

tervention topic $i$, we see if increasing the activation of that feature results in an increase of the feature in the model's output for the ten simple intervention topics. In order to compare across methods, which all have different explanation feature spaces and scales, we measure the success of interventions as a function of the norm of the distance between the edited latent representation $\hat{x}' = g(Edit(f(x)))$ and the original latent representation $x$: $||\hat{x}' - x||/||x||$. Results for intervention success rate are shown in Figure 2 and results for intervened token probability can be found in Appendix B.5.

Across methods and models, we find that by increasing intervention strength, or the magnitude of the edit to the latent representation, intervention success rate first improves and then levels out, as expected. However, we unexpectedly find that Logit lens and Tuned lens generally have the highest intervention success rate, regardless of the normalized edit distance, except when compared to ReFT-r1 on Gemma2-2b. Furthermore, we find that SAEs, probes, and steering vectors require significantly larger edits in order to achieve reasonable intervention success. Note that the minimal edit distance for SAEs is nonzero, as SAE reconstruction incurs a significant error, as explored in Appendix B.1. In general, we believe the lower performance of SAEs is due to heavy noise in the labels of features. For example, a feature labelled 'references to coffee', is sometimes actually a feature that encodes for references to 'beans' and 'coffee beans', and thus only sometimes increases mentions of 'coffee'. Probes and steering vectors also have suboptimal performance, often due to learning of spurious correlations in the training data rather than the true intervention feature.

### 4.3 EFFECTS OF INTERVENTION ON OUTPUT QUALITY

We next measure the coherence of the intervened output text produced by each method to ensure that intervention through interpretability methods is possible without damaging the utility of the model. We measure coherence as described in Section 3 as a function of the intervention success rate in Figure 3 to characterize the tradeoff between intervention success and output coherence. Results for coherence as a function of normalized latent edit distance, $||\hat{x}' - x||/||x||$, are in Appendix B.3. We visualize the mean of coherence scores for the clean model outputs with solid black

Table 1: Correlation between human raters (left) and an LLM rater (Llama3-8b) for coherence or a rules-based grammar checker (right). All three raters are highly correlated with one another.

| | LLM VS HUMAN RATER | | LLM VS ERROR CHECKER | |
|---|---|---|---|---|
| | PEARSON R | $r^2$ | PEARSON R | $r^2$ |
| LLAMA3-8B | 0.94 | 0.75 | -0.96 | 0.92 |
| LLAMA2-7B | 0.80 | 0.68 | -0.85 | 0.73 |
| GEMMA2-2B | 0.80 | 0.67 | -0.78 | 0.75 |
| GPT2-SM | 0.71 | 0.67 | -0.86 | 0.74 |

horizontal lines, the same as those shown in Figure 7, with a buffer of $\pm 1$ around the mean in dashed lines. We also consider a prompting baseline, where we simply prompt the language model to talk about the intervention topic, to better understand the optimal coherence possible while satisfying the intervention. This is shown by the teal stars in Figure3. Prompting was infeasible for GPT2-small as it was not instruction tuned. Also, note that the intervention success rate approaches 100% with prompting as the number of generated tokens increases; however, seeing as we only generate 30 tokens, the success rate may be lower than expected.

Our experiments reveal that while interpretability methods may seem to provide reasonable trade-offs between intervention success and coherence at first glance, they all underperform the simplest baseline of just prompting the model. Furthermore, Logit lens and Tuned lens significantly outperform all other methods when intervening on these simple topics, with intervention success rates of around 0.5 and 0.6 respectively for outputs within one point of deviation from the mean coherence score of the clean model. All other methods exhibit much less desirable Pareto curves, regardless of model size or intervention feature.

**Verifying Coherence.** In order to validate the coherence scores generated through our LLM-as-a-judge setup with Llama3-8b, we verify the coherence scores with human raters. Participants blindly rated 100 outputs for each model, and we measured the correlation between these human ratings and LLM ratings, as shown in Table 1. We find high consistency between both, with particularly high correlation coefficients and $r^2$ values for the larger models.

| Method | Optimal Intervention Strength | Excessive Intervention |
|---|---|---|
| Logit | On my next vacation, I hope to visit the beautiful country of Costa Rica. I've been reading a lot about the country and I'm really excited about the **yoga** retreats I' | My favorite song from the 21st century is "**Yoga**, **Yoga**, **Yoga**" by **Yoga Yoga Yoga Yoga Yoga Yoga Yoga Yoga Yoga Yoga** |
| | (Edit Distance: 0.6, Coherence: 7, Model: Gemma2) | (Edit Distance: 0.7, Coherence: 3, Model: Gemma2) |
| Probing | Think about a time you felt truly at peace. Maybe you were sitting on a beach, or maybe you were sitting in a **yoga** studio. Maybe you were sitting in your own home, or maybe you | My favorite fiction author is the one and only, the one and only, the one and only, the one and only, the one and only, the one and only, the one and only, |
| | (Edit Distance: 0.7, Coherence: 5, Model: Gemma2) | (Edit Distance: 1.1, Coherence: 4, Model: Gemma2) |
| SAE | The coolest wildlife I've ever spotted was a **yoga** class in a tree. I was in the jungle of Costa Rica, and I was doing a **yoga** class in a tree. I was in the jungle of Costa Rica, and I was doing a **yoga** | My favorite museum to visit is the Hat **yoga** practice sequence. It is a sequence that is a sequence that is a sequence that is a sequence that is a sequence that is a sequence that |
| | (Edit Distance: 0.9, Coherence: 6, Model: Gemma2) | (Edit Distance: 1.2, Coherence: 5, Model: Gemma2) |
| Steering | My favorite scientist is Dr. Deepak Chopra. I have been practicing **yoga** for 7 years and I have been practicing meditation for 4 years. I have been practicing **yoga** | Last year, I went to a practice session for a beginner class I was initially initially initially initially initially initially initially initially initially initially |
| | | (Edit Distance: 1.3, Coherence: 5, Model: Gemma2) |
| | (Edit Distance: 0.6, Coherence: 5, Model: Gemma2) | |

Figure 5: Examples of intervened model outputs for intervention feature '**yoga**' at both the optimal intervention strength (left) and the maximum intervention strength tested (left). Outputs degrade into incoherent repetition at high intervention strength for all methods.

We also check the validity of the coherence ratings by comparing with an alternative metric that measures the number of grammatical errors in the intervened output via a rule-based grammar checker. In particular, we use LanguageTool to determine the number of errors in each output, which has thousands of rules relating to grammar, typos, capitalization errors, and more. As expected, there is a high negative correlation between the two, indicating that outputs with more errors are less coherent. However, we note that the number of grammatical errors is not an ideal metric, as it does not assess whether the text generation pertains to the prompt, which an LLM rater can do.

**Qualitative Examples.** We present examples of intervention outputs in Figure 5 for the feature 'yoga,' with more examples in Appendix B.7. We highlight outputs where intervention succeeded with minimal degradation in coherence in "Optimal Intervention Strength" (left column) as well as generations from the highest intervention strength tested (right column) "Excessive Intervention." Note that intervention results in repetition at very high intervention strengths for all methods; however, only Logit Lens and Tuned Lens result in repetition of tokens related to 'yoga.'

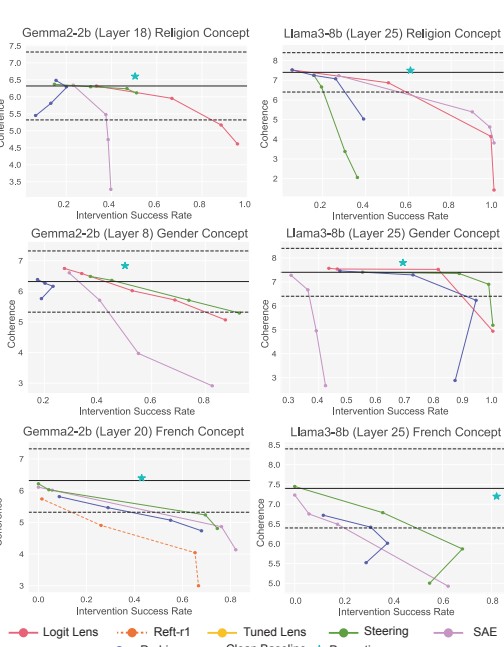

Figure 4: Relationship between intervention success rate and coherence for three complex features: religious references (top), gendered language (middle), and French language (bottom) for Gemma2-2b (left) and Llama3-8b (right).

## 4.4 COMPLEX FEATURES CASE STUDIES

While the aforementioned simple features allow for rigorous evaluation across methods, in practice, users often want to control or steer much more complex concepts. To investigate the feasibility of interventions in more interesting and realistic settings, we present results for three more complex concepts: (1) *religious speech*, with direct or implicit references to a given religion, (2) *gendered-language*, or the ability to preferentially generate text related to a specific gender, and (3) the *French language*, where the model should generate text in French even when given an English prompt. These concepts were chosen from features known to exist in the pretrained and labeled SAEs we evaluated, which had a female gendered-language concept and a Christianity- and Islam-related

religion concept.[2] Intervention success is measured with keyword detectors for the first two concepts and the NLP-based language detector package polyglot (Al-Rfou, 2022) for the French concept.

As shown in Figure 4, we find that even for these more complex concepts, prompting generally outperforms interpretability-based interventions, particularly for the French language concept. Furthermore, no other method performs consistently across models or features, highlighting the unreliability of existing interpretability methods for controlling complex and safety-relevant concepts. However, we generally found that interventions made by Logit lens were most successful in incorporating the intervention into a plausible completion of the prompt. For example, one intervened output was "***Whenever I'm outdoors and in nature, I*** *always have a camera with me. I love taking pictures of God's beautiful creation. I'm a big fan of Jesus and I love spending...*", where we can see that the religion concept is integrated well with the response to the prompt. For most other methods, such as SAEs, interventions either immediately degraded model performance by inducing severe repetition (e.g. "***I had a conversation recently about*** *the last and final prophet of the last and final of the last and the last and the last and*") or they were either unrelated to the prompt or simply incorrect (e.g. "***Tell me an interesting fact about a musical instrument.*** *For example, did you know that the piano is actually a Christian Muslim?*").

### 4.5 INTERVENTION SIMILARITY BETWEEN METHODS

Given that these methods all result in linear edits that should correspond to the same feature, ideally their interventions should all point in the same direction in the model's latent space. We evaluate the empirical similarity between methods by measuring the cosine similarity between *edit directions*, $\hat{x}' - x$, for each intervention topic. The average cosine similarity between these vectors for each intervention topic is shown in Figure 6.

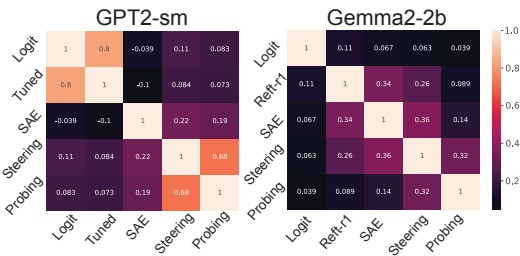

Figure 6: Cosine similarity between methods' intervention directions in model latent space across methods.

We find that Logit Lens and Tuned Lens are highly similar, as expected. Similarly, steering vectors and probe weights tend to lie in similar directions, likely due to the same underlying data used to train both. Most interestingly, we find that sparse autoencoders tend to intervene in somewhat similar directions to steering vectors and probes and have near orthogonal directions to Logit Lens and Tuned Lens, even when interventions succeed for all methods. We speculate that sparse autoencoders may be more similar to probes and steering vectors because the three methods may have a bias toward representing past information and tokens, due to their training and labelling algorithms, also noted by Gur-Arieh et al. (2025). Logit lens and Tuned lens, on the other hand, are designed to reveal information about the *next token* specifically, given that they are early-decoding strategies and thus may contain more information about model outputs rather than inputs.

## 5 CONCLUSION

While interpretability methods show great promise in understanding large language models, the correctness of their explanations is less clear. Do these explanations reveal truth about model computation or simply fool human researchers? We believe that systematic benchmarking of explanations is critical to answer this question. Our work makes progress towards this goal, and answers this question somewhat negatively, showing that current explanations are less accurate than expected. Our work also raises questions regarding the utility of such methods, as we find that prompting outperforms current interpretability methods in its ability to steer models, without requiring any training, data, or access to model weights. We hope future work can address these shortcomings of current methods, paving way toward interpretability methods that are faithful and provide actionable insights for improving and controlling models.

---

[2]ReFT-r1 did not have a feature that directly corresponded to French language, so we consider the closest successful feature available: "French connective and referential pronouns."

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

## A    APPENDIX

## B    APPENDIX

### B.1    ADDITIONAL EVALUATIONS: SANITY CHECKING EXPLANATION RECONSTRUCTIONS

Before testing these methods for their ability to intervene, we first want to evaluate the completeness of the explanations and the effect of replacing $x$ with $\hat{x}$ *without* any intervention or editing. We do so by measuring the normalized latent reconstruction error: `Error` $= ||\hat{x} - x||/||x||$ where $\hat{x} = g(f(x)) = g(z)$. This error is a key part of the loss function that sparse autoencoders are trained on and measures the information loss incurred by mapping between the language model's latent space and the interpretable feature space. Given that steering vectors and linear probes do not output complete explanations, we only measure this error for the other three methods, as shown in Table 2, where we see that errors vary a lot across models but most methods are relatively consistent in their error, with the exception of the GPT2-small sparse autoencoders.

Table 2: Normalized latent reconstruction error without intervention.

| Method | Gemma2-2b | Llama2-7b | GPT2-small |
|--------|-----------|-----------|------------|
| LOGIT LENS | 0.52 | $5e^{-5}$ | 0.22 |
| TUNED LENS | – | $5e^{-3}$ | 0.32 |
| SAEs | 0.38 | – | 1.64 |

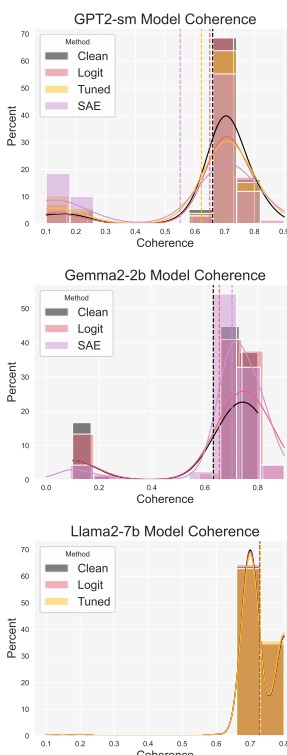

Figure 7: Histogram of coherence scores for clean model outputs (Clean) and for the models where $x$ is replaced by $\hat{x}$ without any intervention for Logit Lens, Tuned Lens, and SAEs. Dashed lines show the mean for each distribution.

## B.2 ADDITIONAL EVALUATIONS: COHERENCE OF METHOD OUTPUTS WITHOUT INTERVENTION

We measure the coherence of the outputs produced by replacing $x$ with $\hat{x}$, as shown in Appendix Figure 7, which we can compare to the baseline of the clean model outputs (labelled 'Clean' and shown in black). We find that the coherence of the outputs generated by the reconstructed latents generally matches the coherence of the clean model outputs. We use a deviation of $\pm 1$ around the mean of clean output coherence scores as a threshold for future evaluations, shown in the dashed lines.

## B.3 ADDITIONAL EVALUATION: COHERENCE OF INTERVENTION WITH RESPECT TO EDIT DISTANCE

We measure the coherence of the intervened output text produced by each method to ensure that intervention through interpretability methods is possible without damaging the utility of the model. We measure coherence as described in Section 3 as a function of normalized latent edit distance,

$||\hat{x}' - x||/||x||$ in Figure 8. We find that even the smallest interventions made with Logit lens and Tuned lens result in significant degradation of model outputs, with a less noticeable dropoff for the other methods. We also plot coherence as a function of the intervention success rate in Figure 3 to characterize the tradeoff between intervention success and output coherence.

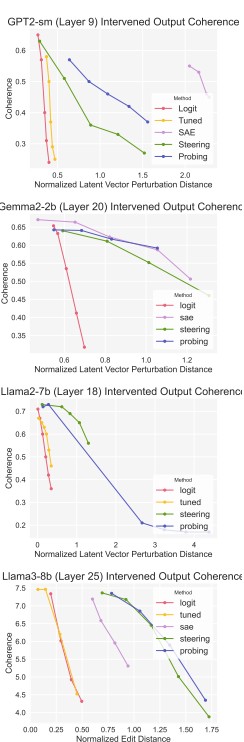

Figure 8: Analysis of coherence of the intervened outputs, measured with Llama3.1-8b, as a measure of the edit distance or magnitude of intervention made. Lens-based methods suffer drastic drops in coherence with only small edits.

### B.4 ADDITIONAL EVALUATIONS: INTERVENTION EFFICACY ACROSS MODEL DEPTH

In order to ensure the generalizability of the above results across layer depths, we repeat all experiments for each layer of GPT2-small, as shown in Figure 9. However, due to some sparse autoencoder features only existing in some layers, we could only consider intervention topics { 'beauty', 'coffee', 'dogs'}. We hold the hyperparameter $\alpha$ that controls for intervention "strength" constant across all layers. Note that this is NOT equivalent to holding the normalized edit distance constant, as shown in the rightmost plot.

We find that layer depth seems to have minimal effect for SAEs and probing, with the exception of the first and last layers. For steering vectors, we observe a modest increase in intervention success rate with increased layer depth and a much more drastic increase in the success rate at later layers for Logit Lens and Tuned Lens. However, as we increase $\alpha$ significantly, we find that the curves for all three methods on intervention rate shift left until the pass rate is approximately 1 at all layers. Intuitively, this makes sense, as any edits to the residual stream at layer 0 will affect the residual stream at later layers. We note that these results highlight the need to tune the intervention strength for each method, each model, and each layer - limiting their ease of use.

### B.5 ADDITIONAL METRICS: INTERVENED TOKEN PROBABILITY

Please see Section 3.2 for more details. We measure the probability assigned to tokens relating to feature $i$ when intervening on feature $i$. As such, even if a model's output does not directly reflect interventions made to $z_i'$ due to sampling, we can measure if increasing the activation of feature $i$

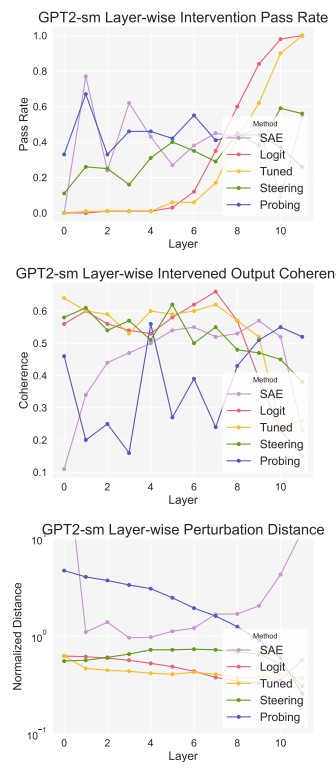

Figure 9: Analysis of intervention pass rate (left), coherence (middle) and edit distance (right) across all layers of GPT2-sm. We find that intervening at later layers is significantly more effective for Logit and Tuned Lens than earlier interventions, but probes, steering vectors, and SAEs are relatively invariant to the choice of layer.

results in any change to the model's output at all. We refer to this metric as **Intervened Token Probability**.

Results for Intervened Token Probability are shown in Figure 10, where we see that intervention with all methods across all models increases the probability of intervention-related tokens, even if the intervention does not succeed. We also note that there is a significant difference between the order of magnitude of the intervened token probability for sparse autoencoders, around $10e^{-5}$ and the rest of the methods, which range from $10e^{-4}$ to 0.5.

## B.6 ADDITIONAL METRICS: PERPLEXITY

As described in Section 3.2, we evaluate the perplexity of the intervened generated text to measure the utility of interpretability methods for targeted intervention in 11. We measure this perplexity with respect to a stronger language model than the one studied, in this case with Llama3.1-8b.

We find that the results for perplexity are generally unintuitive and do not align with the results for coherence. We hypothesize that perplexity is not a useful measure when text is extremely out-of-distribution with respect to normal text, and in particular when the text is highly repetitive. For example, if the same token is repeated 20 times, we (and other language models) might assume that the next 20 tokens would also be the same, resulting in a low perplexity even if the quality of the text is poor. As such, we do not consider these results to be particularly meaningful or significant.

## B.7 ADDITIONAL EXAMPLE OUTPUTS

We present additional examples of the output text for all intervention methods in Figures 12 and 13 for qualitative evaluation of intervention on the feature 'coffee' and 'San Francisco'. Examples for the "Optimal intervention strength" (left column) were randomly chosen from the outputs where in-

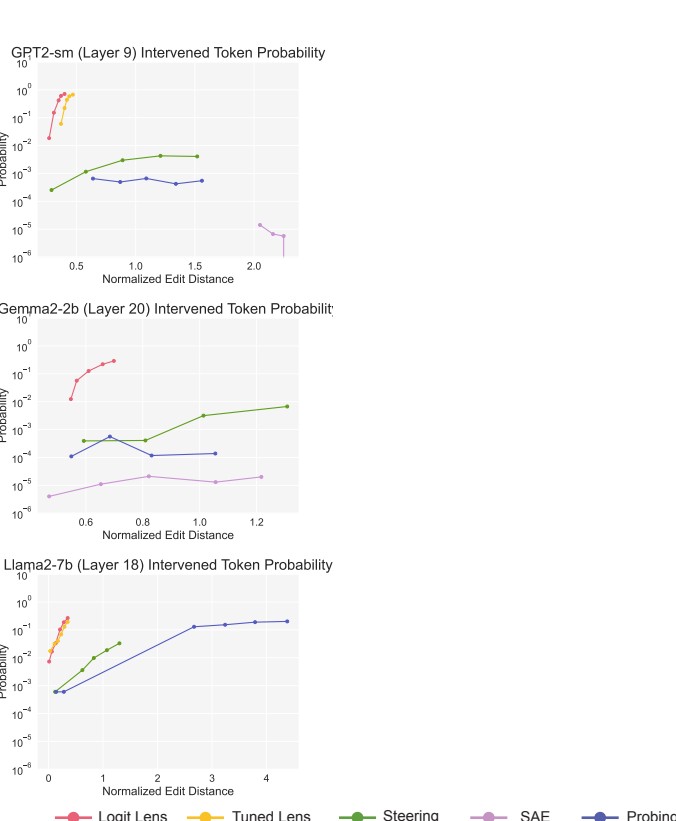

Figure 10: Evaluation of intervention success with respect to the probabilities of the tokens corresponding to the features intervened on for each method. Note that normalized edit distance is a proxy for intervention intensity that is comparable across methods.

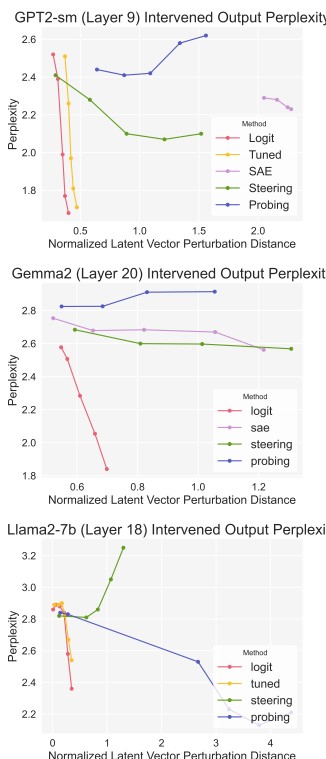

Figure 11: Analysis of perplexity of the intervened outputs, measured with Llama3.1-8b, as an alternative metric to Coherence. We find that perplexity does not align with Coherence, as highly repetitive sequences may have low perplexity despite being incoherent answers to prompts.

tervention succeeded and coherence was still relatively high. Examples for "Excessive Intervention" were randomly chosen from the outputs of the highest intervention strength tested (right column). Please see Section 4.3 for more.

### B.8 IMPLEMENTATION DETAILS: OPEN-ENDED GENERATION

In order to generate open-ended text after intervening on the explanation, we edit the corresponding representations *in place*, as is common practice with prior steering methods. Formally, the representation $x_t$ at token position $t$ and layer $l$ is edited to be $\hat{x}_t'$, ensuring a causal effect on all ensuing tokens $x_{t+1}, x_{t+2}, ..., x_T$.

### B.9 IMPLEMENTATION DETAILS: INTERVENTION HYPERPARAMETER $\alpha$

When intervening on $z$ to get $z'$ with Logit Lens, Tuned Lens, and SAEs, we set $z_i' = \alpha * max(z)$. For probing and steering vectors, $\hat{x}' = x + \alpha * v$, where $v$ is the steering vector or the weights of the linear probe. Note that $\alpha$ is a hyperparameter that must be tuned for each method and model, and thus cannot be used to compare the effects of interventions across methods. We record the values of $\alpha$ used in our experiments in Table 3.

### B.10 IMPLEMENTATION DETAILS: SAE FEATURES

As described in Section 4.1, we use the sparse autoencoders hosted on SAELens and find the relevant features with Neuronpedia's exploration and search tools. We document all of the features we consider for each intervention topic in Table 4. Note that for some specified intervention topics, an exact feature match does not exist for the GPT or Gemma SAEs. As such, we either exclude that topic or consider the closest-related topic (such as "instruction related to yoga poses and their

| Method | Optimal Intervention Strength | Excessive Intervention |
|---|---|---|
| Logit | In my daydreams, I often find myself in a **coffee** shop, sipping a cup of **coffee** and reading a book. I love the smell of **coffee** and the taste of **coffee**.

*(Edit Distance: 0.6, Coherence: 6, Model: Gemma2)* | The joys and challenges of traveling alone include **coffee** shops, museums, **coffee** shops, **coffee** shops, **coffee** shops, **coffee** shops, **coffee** shops, **coffee** shops, **coffee** shops, **coffee** shops, **coffee**

*(Edit Distance: 0.7, Coherence: 2, Model: Gemma2)* |
| Probing | I have travelled far and wide, and, I have seen many different types of **coffee**. I have had the pleasure of trying some of the best **coffee** in the world. I have also had the

*(Edit Distance: 0.8, Coherence: 5, Model: Gemma2)* | Whenever I'm outdoors and in nature, I '41111111111111111111111111111

*(Edit Distance: 1.1, Coherence: 1, Model: Gemma2)* |
| SAE | I plan to write my Ph.D. dissertation on the topic of the relationship between the **coffee** and the **coffee** shop. I am interested in the history of the **coffee** shop, the history of the **coffee** shop

*(Edit Distance: 0.9, Coherence: 5, Model: Gemma2)* | In my previous life, I'm sure that I was a **coffee** shop barista. I' **coffee** is always fresh, the **coffee** is always fresh, and the **coffee** is always fresh. I's a great

*(Edit Distance: 1.2, Coherence: 3, Model: Gemma2)* |
| Steering | List some fun facts to share with kids.
 * The first **coffee** shop in the world was in 1977 in the city of Seattle, in the state of Washington.

*(Edit Distance: 0.6, Coherence: 7, Model: Gemma2)* | My favorite national park to visit is:  the National Park: The 

*(Edit Distance: 1.3, Coherence: 1, Model: Gemma2)* |
| Tuned | Check out this haiku I wrote:
**coffee**, my dear friend
warming my hands and heart
in this cold world

*(Edit Distance: 0.4, Coherence: 6, Model: Llama2)* | The coolest wildlife I've ever spotted was a black **coffee coffee coffee coffee coffee coffee coffee coffee coffee coffee coffee coffee coffee coffee coffee coffee coffee coffee coffee coffee coffee coffee coffee coffee coffee coffee coffee coffee coffee coffee coffee**

*(Edit Distance: 0.5, Coherence: 2, Model: Llama2)* |

Figure 12: Example outputs with intervention on "coffee" feature.

| Method | Optimal Intervention Strength | Excessive Intervention |
|---|---|---|
| Logit | One dramatic impact of climate change on wildlife will be the spread of disease. The **San Francisco** Zoo is taking steps to prepare for the arrival of the first cases of disease in the San Francisco Bay Area.

*(Edit Distance: 0.6, Coherence: 8, Model: Gemma2)* | A book on art history that I found fascinating is Francisco Goya, Francisco Goya Francisco Goya Francisco Goya Francisco Goya Francisco Francisco Francisco Francisco Francisco Francisco Francisco Francisco Francisco Francisco Francisco Francisco

*(Edit Distance: 0.7, Coherence: 2, Model: Gemma2)* |
| Probing | Next month, I plan to travel to the United States to attend the 2016 International Conference on the History of the Book. The conference will be held in **San Francisco**, California

*(Edit Distance: 0.7, Coherence: 8, Model: Gemma2)* | A book on art history that I found fascinating is The Modern Art Show: The 1932/33 Art/Alfar/Alfar/Alfar/

*(Edit Distance: 1.1, Coherence: 1, Model: Gemma2)* |
| SAE | During my last work trip, I was able to visit the **San Francisco** Museum of Modern, and I was able to see the new exhibit, "The Last Resistance." The exhibit is a

*(Edit Distance: 0.9, Coherence: 8, Model: Gemma2)* | Some examples of eco-friendly destinations and tips for minimizing your travel footprint are San Marin Island, Tib Travel, The Golden Emb Emb Emb

*(Edit Distance: 1.2, Coherence: 1, Model: Gemma2)* |
| Steering | My favorite childhood memory is of my parents taking me to the **San Francisco** Exploratorium. I remember being amazed by the Exploratorium's Exploratory Playroom, which was a

*(Edit Distance: 0.6, Coherence: ,8 Model: Gemma2)* | In my daydreams, I like to imagine that I'm a 19th-sf Francisco Francisco Francisco Francisco Francisco Francisco Francisco Francisco Francisco Francisco Francisco Francisco Francisco Francisco

*(Edit Distance: 1.3, Coherence: 2, Model: Gemma2)* |
| Tuned | Check out this haiku I wrote:
 **San Francisco**'s hills
Steep and winding, a challenge
To walkers, bikers too

*(Edit Distance: 0.4, Coherence: 7, Model: Gemma2)* | My favorite song from the 21st century is "Ho Hey" by **San Francisco**-based indie rock band The **San Francisco** Francisco Francisco Francisco Francisco Francisco Francisco Francisco Francisco

*(Edit Distance: 0.5, Coherence: 3, Model: Llama2)* |

Figure 13: Example outputs with intervention on "San Francisco" feature.

benefits" when what we would like is "references to yoga"). Many of these imperfect features still yield reasonable intervention success rates.

Table 3: Values for hyperparameter $\alpha$ used to control intervention edit distance for each method and model.

| Method | GPT2-small Layer 9 | Gemma2-2b Layer 20 | Llama2-7b Layer 18 |
|---|---|---|---|
| Logit Lens | [50, 70, 90, 110, 130] | [100, 130, 160, 200, 230] | [0.5, 3, 7, 11, 15, 19] |
| Tuned Lens | [20, 25, 30, 35, 40] | – | [1, 7, 11, 15, 19, 23] |
| SAEs | [3, 4, 5, 6] | [1, 2, 3, 4, 5] | – |
| Probing | [150, 200, 250, 300, 350] | [200, 250, 300, 350] | [10, 90, 110, 130, 150] |
| Steering Vectors | [2, 4, 6, 8, 10] | [3, 4, 5, 6] | [0.5, 3, 4, 5, 6] |

Table 4: Specific SAE features used for intervention on GPT2-sm and Gemma2-2b. The feature ids and their according Neuronpedia labels are provided.

| Intervention Feature | GPT2-small Layer 9 Feature | GPT2-small SAE Layer 9 Name | Gemma2-2b Layer 20 Feature | Gemma2-2b SAE Layer 20 Feature Label |
|---|---|---|---|---|
| San Francisco | 11233 | "mentions of the city of San Francisco" | 3124 | "references to San Francisco and related locations" |
| New York | 5831 | "references to the city of New York" | 3761 | "specific place names and geographical locations in New York" |
| beauty | 1805 | "words related to beauty or aesthetic appreciation" | 485 | "instances of the word "beauty" in various contexts" |
| football | – | – | 11252 | "references to football and baseball contexts" |
| pink | 2415 | "mentions of the word "Pink."" | 13703 | "references to the color pink and its various associations" |
| dogs | 12435 | "mentions of dogs or dog-related terms" | 12082 | "references to dog behavior and interactions" |
| yoga | – | – | 6310 | "instructions related to yoga poses and their benefits" |
| chess | 21685 | "mentions of the game of chess" | 13419 | "elements within the context of chess" |
| snow | 5053 | "references to snow-related terms" | 13267 | "references to snow and related terms" |
| coffee | 23472 | "references to coffee-related words" | 15907 | "references to coffee and related cafés or establishments" |