# OpenReview forum: "Towards Unifying Interpretability and Control: Evaluation via Intervention"
_ICLR.cc/2025/Workshop/BuildingTrust — Submitted to BuildingTrust_

### Official Review · Reviewer_SMZz · 2025-02-21
**Ambitious goal for benchmarking interpretability methods through interventions, makes a good stride towards this goal. Experiments are compelling, but could target more abstract interventions.**

**Rating:** 6
**Confidence:** 3

**Review:**

### Strengths
- This paper sets an ambitious and essential goal for benchmarking and evaluating interpretability research through interventions. A central difficulty in interpretability research is quantifying the effectiveness of interpretability technique, and this work makes significant strides in this direction.
- Experiments are extensive and well-explained. The use of human evaluators to validate the large-scale use of LLM checkers was a strong inclusion. I also particularly enjoyed the inclusion of *Intervention Similarity Between Methods*, to better understand the commonalities between intervention methods.

### Weaknesses
- Instead of exploring the listed 10 intervention topics, I found the most compelling evaluation to be in Section 4.4 Complex features. These evaluations are more in line with the stated goals of improving safety and debiasing. In my opinion, interventions on complex features of these sorts should have been the primary focus of the experiments/benchmarks.
- I am not certain what to take away from this paper in terms of comparison of intervention methods. I think the paper's conclusions could have been expanded on.
- (Minor) There are a few formatting choices in this paper that I found slightly distracting. I've included some suggested changes below to improve clarity.

### Suggestions
- I had difficulties understanding Figure 1 when I first encountered it, as I found it a bit overwhelming. I suggest that the table from Figure 1 be moved to Section 3.
- I found Figures 2 and 3 to be a bit too small. If kept at their current dimensions, I would suggest increasing the thickness of the lines.
- The **Interventions Topics** paragraph (lines 290-300) could be rewritten for better clarity.
- Llama3.1-8b is listed as the evaluator on lines 256 and 258, but the caption for Table 1 lists Llama3-8b
- Appendix header is listed twice.
- Appendix figures should be reformatted to better fill the pages.

---

### Official Review · Reviewer_k4h2 · 2025-03-02
**Important direction, but the specific causal intervention evaluation does not provide a fair or accurate comparison of interpretability methods**

**Rating:** 5
**Confidence:** 4

**Review:**

# Summary
This paper compares several interpretability methods (sparse autoencoders, Logit Lens, Tuned Lens, and probing) by studying their effectiveness as causal intervention tools. The authors evaluate interventions by testing how often the model produces certain tokens in the answer to an open-ended question. They study the tradeoff between intervention effectiveness and output coherence. Across several model sizes, they find that logit-based interventions are most effective in producing specified tokens, though all methods generally underperform simple prompting baselines.

# Strengths
- The paper is clearly written with a well-structured approach to comparing interpretability methods.
- The problem is well-motivated, addressing an important gap in how to evaluate and compare interpretability techniques.
- The unified encoder-decoder framework provides a useful abstraction for comparing methods with different underlying mechanisms.
- The coherence-intervention trade-off analysis provides valuable insights about the practical utility of these methods.

# Weaknesses
- The definition of feature explanation “correctness” lacks nuance. The authors define correctness in terms of effect on the output. While some interpretability methods are designed to produce an effect on the output, others can be useful for providing insights about internal representations.
- While it is valuable to evaluate different interpretability methods based on their downstream effects, the evaluation should be put in the context of a downstream task. In this project, the downstream task is to output a certain token. This task naturally favors Logit Lens, which directly manipulates token probabilities. Other intervention types like SAEs or probes might represent high-level concepts or be less specific to a certain token, so their effectiveness should be measured on different types of downstream tasks. Although there is also some analysis of high-level concepts, their analysis is not as thorough.
- The paper is missing some important implementation details, like which SAE width they are using for the features listed. There is also no mention of how model generations are sampled (temperature, top-k, etc.), which significantly impacts both the success rate of interventions and output coherence measurements.
- The SAE features used are not very specific to the concepts they are evaluating them for (if I inferred correctly that they were using the 16k SAEs for Gemma 2 2b, which is not specifed). For example, the San Francisco feature for Gemma 2 2b activates on locations in the Bay Area in general and not only San Francisco, and similarly for the New York feature. This makes their evaluation metric hard to interpret when comparing different intervention methods.
- The SAE error should probably be added back to the decoded x’, although this doesn’t seem to have a huge effect on the output and this effect is analyzed in the appendix.
The paper would benefit from more discussion of why prompting outperforms the mechanistic methods tested. While not surprising, it calls into question the effectiveness of these techniques at steering model outputs.

---

### Official Review · Reviewer_qRdX · 2025-03-02
**Review for "Towards Unifying Interpretability and Control: Evaluation via Intervention"**

**Rating:** 6
**Confidence:** 4

**Review:**

**Summary**

This paper investigates the relationship between interpretability and control in large language models (LLMs), proposing that effective interpretability should enable precise interventions. The authors adapt four interpretability methods—Logit Lens, Tuned Lens, Sparse Autoencoders (SAEs), and Probing—for use in intervention scenarios. They introduce a unified encoder-decoder framework to integrate these methods and assess their effectiveness in enabling controlled modifications. Two new metrics, Intervention Success Rate (ISR) and Coherence-Intervention Tradeoff (CIT), are proposed to evaluate whether interpretability methods allow for controlled changes without compromising coherence. Experiments on open-weight models reveal that lens-based methods outperform SAEs and probing for simple interventions but often degrade coherence.

**Strengths**

- **Adaptation of interpretability methods for intervention**: The authors put significant effort into adapting lens-based methods for intervention, such as finding pseudo-inverses, which is a notable contribution.
- **Fair comparison of methods**: The introduction of metrics like "normalized edit distance" ensures a fair comparison across interpretability methods. Such benchmarks could greatly benefit the interpretability research community.

**Weaknesses**

- **Unfair comparison for SAEs**: The comparison may be biased against SAEs. The paper identifies SAE features for a concept by searching in Neuronpedia. Neuronpedia explains SAE features by analyzing the text that activates them using LLMs but does not consider their causal effects. Recent work, such as Chalnev et al. (2024), has improved steering with SAEs, and Paulo et al. (2024) have proposed a method to explain SAE features while accounting for their causal effects. These approaches should theoretically enhance the performance of SAEs for steering but are not considered in the paper, potentially skewing the results.
- **Questionable premise**: The claim that "effective interpretability should enable precise interventions" is debatable. Methods like SAEs were originally designed for understanding model internals, not for intervention. This premise may not align with the goals of all interpretability methods.

**References**

- Chalnev, S., Siu, M., & Conmy, A. (2024). Improving Steering Vectors by Targeting Sparse Autoencoder Features. *arXiv preprint arXiv:2411.02193*.
- Paulo, G., Mallen, A., Juang, C., & Belrose, N. (2024). Automatically Interpreting Millions of Features in Large Language Models. *arXiv preprint arXiv:2410.13928*.

---

### Decision · Program_Chairs · 2025-03-04

**Decision:**

Reject

**Comment:**

The reviewers generally find the paper well-motivated, clearly structured, and a valuable contribution to evaluating interpretability methods through interventions. However, concerns remain regarding the fairness of comparisons, the premise that interpretability should always enable control, and the evaluation setup favoring certain methods (e.g., Logit Lens). Additionally, more discussion on why prompting outperforms mechanistic methods and greater focus on complex feature interventions would strengthen the impact.